# Increased Prevalence of Neuropsychiatric Disorders during COVID-19 Pandemic in People Needing a Non-Deferrable Neurological Evaluation

**DOI:** 10.3390/jcm10215169

**Published:** 2021-11-04

**Authors:** Giacomo Tondo, Davide Aprile, Fabiana Tesser, Cristoforo Comi

**Affiliations:** 1Neurology Unit, S. Andrea Hospital, Department of Translational Medicine, University of Piemonte Orientale, Corso Abbiate 21, 13100 Vercelli, Italy; giacomo.tondo85@gmail.com (G.T.); fabiana.tesser@aslvc.piemonte.it (F.T.); 2School of Psychology, Vita-Salute San Raffaele University, 20132 Milan, Italy; 3Department of Translational Medicine, University of Piemonte Orientale, 28100 Novara, Italy; 20017183@studenti.uniupo.it; 4Interdisciplinary Research Center of Autoimmune Diseases (IRCAD), University of Piemonte Orientale, 28100 Novara, Italy

**Keywords:** neuropsychiatric symptoms, behavioral symptoms, psychological symptoms, quarantine, coronavirus disease, neurology unit

## Abstract

Background: The novel coronavirus disease of 2019 (COVID-19) outbreak provoked a profound healthcare system reorganization. This study aimed to compare the reasons for requesting a non-deferrable neurological evaluation before the COVID-19 pandemic and during the lockdown. Methods: Retrospective observational study including non-deferrable neurological outpatients before the pandemic (pre-COVID-19 group, *n* = 223) and during the Italian second wave of the COVID-19 pandemic (LOCKDOWN group, *n* = 318). Results: The number of patients sent for cerebrovascular disorders, headache, and vertigo significantly dropped between the pre-COVID-19 era and the lockdown period. While in the pre-COVID-19 group, the most frequent diagnosis was cerebrovascular disorder; neuropsychiatric disorders ranked first in the LOCKDOWN group. Moreover, the percentage of appropriate non-deferrable neurological evaluations significantly increased in the LOCKDOWN group compared with the pre-COVID-19 group. Discussion: Our study shows a significant increase of neuropsychiatric disorders in non-deferrable neurologic evaluations during the Italian second wave of the COVID-19. Overall, cases were more severe and required a more complex management during the lockdown compared with the pre-COVID era. These findings confirm that a careful approach to prevent the psychological consequences of the pandemic is needed, and long-term rearrangements of the healthcare system are desirable to guarantee appropriate management.

## 1. Introduction

The novel coronavirus disease of 2019 (COVID-19) outbreak caused a profound healthcare reorganization worldwide. In responding to the growing number of COVID-19 patients, healthcare worker activity has been redirected to COVID-19 assistance [1]. Several hospitals and departments have been converted to COVID-19 care centers, and new critical units have been equipped to face the rapid pandemic spread [2]. Since the first weeks of the pandemic, the reallocation of healthcare workers and resources provoked the closure of non-urgent outpatient clinics. Only non-deferrable consultations were preserved in most hospitals. In addition, the infection control measures induced critical changes in healthcare service delivery and in access to hospitals by patients worried about being infected or about not receiving adequate care. This healthcare system reorganization led to dramatic consequences, especially in neurological patients affected by chronic disorders needing regular follow-up.

Worldwide, patients affected by chronic neurological disorders experienced an increased burden of neurological symptoms and faster deterioration, with devastating psychological consequences [3,4,5,6]. Worsening of preexisting cognitive and neuropsychiatric disturbances was reported over the lockdown [7,8]. Italy was the first European country that officially reported a death due to COVID-19 in February 2020 and it was dramatically affected by the virus spread and related measures during the first wave [9]. The Italian government implemented urgent measures to limit the virus’ diffusion, which were extended from the regional to the whole national area on 9 March 2020 [10]. From May to September 2020, there was a temporary release of lockdown constraints, but in October 2020, Italy faced the second COVID-19 wave, which had already hit Europe again [11]. The reorganization of the healthcare system in response to the pandemic spread and its effects on outpatient clinical activity has been analyzed in several departments [12,13,14,15]. A recent study on a large Chinese population investigated the impact of the prolonged lockdown measures on healthcare organization in the neurology outpatient clinic, reporting a significant drop in patients’ access to the hospital and a significant increase of neuropsychiatric disturbances [16]. Despite profound changes in ordinary clinical activity, with the closure of non-urgent visits, non-deferrable neurological evaluations were guaranteed in most hospitals in several countries, even during the second wave. The effects of the lockdown on the access and management of patients sent for a non-deferrable neurological evaluation have not been investigated yet. 

In the current study, we aimed to investigate similarities and differences in requesting a neurological non-deferrable evaluation, in diagnosis, and in prescribed therapy before the pandemic period and during the lockdown. In addition, we employed a previously used definition of avoidable visit (a visit that did not require any diagnostic test, medical procedure, or medication change) [17] to compare the number of non-avoidable evaluations before and after the COVID-19 pandemic, to reveal potential changes in the clinical management of neurological patients. 

## 2. Materials and Methods

This retrospective observational study was performed at the Sant’Andrea Hospital in Vercelli, Piedmont, Italy. Data were extracted from the electronic health record system of the neurology unit. We included outpatient non-deferrable neurological visits between 1 October 2019 and 7 March 2020 (pre-COVID-19 group) and between 1 October 2020 and 15 March 2021 (Italian second wave of the COVID-19 pandemic, LOCKDOWN group). We collected all available clinical data, including reason for requesting a non-deferrable neurological evaluation, main diagnosis at discharge, comorbidities, actual therapy, added therapy at discharge, recommended further investigations, medical procedures (including intravenous therapy administration, i.e., steroids or immunoglobulins), and hospitalization. The following reasons for requesting a non-deferrable neurological evaluation were considered: cognitive decline; cerebrovascular disorders (CVDs), including ischemic and hemorrhagic strokes and transient ischemic attacks; epilepsy; headache; Multiple Sclerosis (MS); neuro-muscular (NM) disorders; neuropsychiatric disorders (NPSs), including anxiety, depression, insomnia and psychosis; pain, including limb pain, cervicobrachial pain, low back pain; Parkinson’s disease (PD) and other movement disorders, including tremor and other involuntary movements; transient alteration of consciousness; vertigo; other unclassifiable reasons. The following diagnoses at discharge were considered: cognitive decline, including mild cognitive impairment, neurodegenerative and non-neurodegenerative dementias; CVDs; epilepsy; primary headache; MS; NM disorders, including amyotrophic lateral sclerosis, neuropathies, and Myasthenia Gravis; NPSs; PD, essential tremor (ET), atypical parkinsonism, Huntington’s disease and dystonia; other neurological diagnoses; non-neurological diagnoses; absent diagnosis. 

Since the study involved only non-deferrable neurological evaluations, we aimed to compare the appropriateness of the evaluations before and after the COVID-19 outbreak. However, avoidable, non-urgent, or inappropriate definitions are challenging, lacking a universal consensus and a standard definition [18]. A recent retrospective study investigating the rate of non-urgent visits in the emergency department in the United States used a very conservative definition of avoidable visits, including visits not requiring any diagnostic tests at the discharge, no further procedures or drugs [17]. We applied the same classification to compare avoidable and non-avoidable evaluations in the pre-COVID-19 group and the LOCKDOWN group to underline possible differences in neurological patient management.

The study was performed in compliance with the Declaration of Helsinki. Prospective informed consent was waived due to the retrospective design of the study and the fact that pseudonymized data was used.

All statistical analyses were performed using the IBM Statistical Package for the Social Sciences (SPSS) for Windows, Version 25.0 (IBM Corp., Armonk, NY, USA), with *p* values < 0.05 considered as statistically significant. Continuous variables are presented as mean and standard deviation (SD), while categorical variables are summarized as actual number and percentages. Differences in patient demographics, clinical characteristics and all considered variables were compared using χ^2^ and *t* tests. The prevalence of each diagnosis was compared between the pre-COVID-19 and the LOCKDOWN groups. We especially focused on the diagnoses of NPSs, including anxiety, depression, insomnia, and psychosis. In addition, the number of prescribed therapies at discharge, including antidepressant, antiepileptic, antipsychotic drugs, benzodiazepines, painkillers and other medications, was compared between groups.

## 3. Results

### 3.1. Demographics and Clinical Features

A total of 223 visits in the pre-COVID-19 group and 318 in the LOCKDOWN group were analyzed. Patients’ demographics are listed in Table 1. The mean age (SD) was 62.8 years (17.4) in the pre-COVID-19 group and 60.6 years (17.3) in the LOCKDOWN group. The number of patients at the first evaluation was similar between the two groups (*n* = 106; 48% in the pre-COVID-19 and *n* = 130; 41% in the LOCKDOWN, *p* = 0.125). The percentage of visits which were follow-up evaluations, that is patients previously evaluated at the same neurology outpatient clinic, was 52% in the pre-COVID-19 group and 59% in the LOCKDOWN group. The most frequent reported comorbidities in the pre-COVID-19 group were hypertension (*n* = 123; 39%), hypercholesterolemia (*n* = 54; 17%), diabetes mellitus (*n* = 42; 13%), malignancy (*n* = 38; 12%), hypothyroidism (*n* = 32; 10%), and chronic obstructive pulmonary disease (*n* = 30; 9%). Similarly, in the LOCKDOWN group, patients presented the following comorbidities: hypertension (*n* = 101; 45%); hypercholesterolemia (*n* = 44; 20%); diabetes mellitus (*n* = 27; 12%); malignancy (*n* = 27; 12%); hypothyroidism (*n* = 22; 10%); chronic obstructive pulmonary disease (*n* = 17; 8%). No differences between groups were found regarding the prevalence of each comorbidity. 

Figure 1 details sample selection, patients’ access and reasons for requesting the non-deferrable neurological evaluation.

### 3.2. Differences in Reasons for Requesting a Non-Deferrable Neurological Evaluation

In the pre-COVID-19 group, the most frequent reasons for requesting a non-deferrable neurological evaluation were CVDs (*n* = 58; 26%), headache (*n* = 37; 17%), and epilepsy (*n* = 26; 12%). In the LOCKDOWN group, a non-deferrable neurological evaluation was requested primarily due to PD or other movement disorders (*n* = 53; 17%), followed by CVDs (*n* = 38; 12%), headache (*n* = 34; 11%), and NM disorders (*n* = 34; 11%). The number of patients sent to the neurologist due to CVDs, headache, and vertigo was significantly reduced over the lockdown period, while requests for MS, PD and other movement disorders and transient alteration of consciousness significantly increased over the lockdown period compared to the pre-COVID-19 era (Table 1).

### 3.3. Differences in the Prevalence of Diagnoses at Discharge

Several statistical differences were observed in the prevalence of diagnoses at discharge. In the pre-COVID-19 group, the most frequent diagnoses were CVDs (*n* = 49; 22%), primary headache (*n* = 35; 16%), epilepsy (*n* = 29; 13%), and NM disorders, including neuropathy (*n* = 17), myasthenia gravis (*n* = 10), and amyotrophic lateral sclerosis (*n* = 2). In the LOCKDOWN group, NPSs, including depression, anxiety, insomnia, and psychosis, were the main diagnosis (*n* = 57; 18% of cases), followed by a neuromuscular disorder (*n* = 44; 14%) and PD, and other movement disorders (*n* = 38; 12%). Specifically, in the LOCKDOWN group 5 patients had isolated depression, 1 patient had isolated anxiety, 1 patient had insomnia, 1 patient had psychosis, and 49 patients were diagnosed with two or more NPSs. Within the NM disorders diagnoses, 28 patients were diagnosed with neuropathy, 14 patients were diagnosed with myasthenia gravis, and 2 patients had a diagnosis at discharge of amyotrophic lateral sclerosis. Among patients diagnosed with movement disorders, 18 were diagnosed with PD (4 patients received a PD diagnosis in the pre-COVID-19 group), 9 patients were diagnosed with ET (6 in the pre-COVID-19 group), 7 patients had a diagnosis at discharge of atypical parkinsonism (3 in the pre-COVID-19 group), 3 patients had dystonia, and 1 patient had Huntington’s disease (no patients with dystonia or chorea were evaluated in the pre-COVID-19 group). When comparing the pre-COVID-19 and the LOCKDOWN group, significant differences were found in the prevalence of diagnoses of CVDs and primary headache, which were decreased in the lockdown period, and in the diagnoses of NPSs, MS, PD and other movement disorders, which were significantly increased in the LOCKDOWN group. 

### 3.4. Appropriateness of the Non-Deferrable Neurological Evaluations

According to the used classification system, we considered “avoidable” visits those visits that ended without any changes in therapy, without further medical procedures and without recommended diagnostic tests. In the pre-COVID-19 group, *n* = 69 (31%) visits were considered avoidable and *n* = 154 (69%) were considered non-avoidable. In the LOCKDOWN group, *n* = 63 (20%) visits were classified as avoidable, and *n* = 255 (80%) were classified as non-avoidable. Thus, in the LOCKDOWN group, non-avoidable visits were significantly increased compared to the pre-COVID-19 group (*p* = 0.003). Interestingly, in the whole pre-COVID-19 group, 7 patients (3%) required hospitalization, while in the LOCKDOWN group, 21 (7%) patients required hospitalization (*p* = 0.073). 

### 3.5. Subanalysis of NPSs and Prescribed Therapy

As a further analysis, we explored differences between groups regarding diagnoses of specific NPSs and the prescribed therapies. Results showed a significantly higher prevalence of depression (χ^2^ = 10.49; df = 1; *p* = 0.001) in the LOCKDOWN group compared with the pre-COVID-19 group. Overall, the number of prescriptions, including antidepressant, antiepileptic, antipsychotic drugs, benzodiazepines, and painkillers, was lower in the pre-COVID-19 compared with the LOCKDOWN group (χ^2^ = 11.17; df = 1; *p* < 0.001). When exploring differences in the subgroups of prescribed drugs, a significant increase in antidepressant therapy prescriptions emerged in the LOCKDOWN group (χ^2^ = 4.67; df = 1; *p* = 0.030) (Figure 2).

## 4. Discussion

The present study shows significant changes in the reasons for access and for primary diagnoses in a non-deferrable neurology outpatient clinic during the Italian second wave of the COVID-19 spread. While in the pre-COVID-19 era non-deferrable neurological evaluations were conducted primarily due to CVDs, during the lockdown, the principal diagnosis was, in the majority of cases, a neuropsychiatric disturbance, including depression, anxiety, insomnia, and psychosis. Overall, the admitted patients during the lockdown required further investigation or medical procedures or novel drugs in a higher percentage of cases. 

The COVID-19 pandemic provoked profound changes in every aspect of our lives. Worldwide, access to the healthcare system has changed in response to the spread of the virus. To reduce the risk of contagion and virus diffusion, most hospitals underwent a temporary closure of routine activities, maintaining only non-deferrable evaluations. In addition, the needs and reasons for being admitted to hospital have changed. 

Our study shows significant changes in the reasons for requesting a non-deferrable neurological evaluation during the lockdown. In the pre-COVID-19 era, non-deferrable neurological evaluations were requested mainly due to CVDs and headache. During the lockdown, the number of patients complaining of CVDs and headache dropped significantly, together with the visits requested due to vertigo. Similar findings, namely a significant reduction in the proportion of patients complaining of headache, dizziness, and CVDs from 2019 to 2020, were reported in a recent study analyzing clinical activity changes in a Chinese neurology outpatient clinic [16]. A global reduction in the evaluation of CVDs and in the treatment and hospitalization of stroke and transient ischemic attacks has been reported in Italy and in different countries [19,20,21]. Social distancing and subsequent isolation may have caused the decreased stroke detection or these may have influenced patients to avoid seeking medical support due to fear of being infected [22,23]. The fear of the virus, associated with concerns about not receiving adequate medical assistance, and the willingness to not represent a further burden for an overwhelmed healthcare system, may also explain the drop in visits requested for minor or mild symptomatology, including headache and vertigo [16,22]. 

Conversely, visits due to a challenging diagnosis, including transient alteration of consciousness, and neurological disorders requiring complex therapeutic management, such as MS, PD, and movement disorders, significantly increased over the lockdown. This aspect was particularly evident in patients diagnosed with movement disorders. We observed that PD patients evaluated at the non-deferrable neurology clinic tripled during the lockdown (2% in the pre-COVID-19 group vs. 6% in the LOCKDOWN group). Similarly, patients diagnosed with atypical parkinsonism doubled during the lockdown period (1% in the pre-COVID-19 group vs. 2% in the LOCKDOWN group). Lastly, patients with dystonia or Huntington’s disease were forced to access the non-deferrable clinic over the lockdown (no patients with these characteristics were admitted in the pre-COVID-19 group). This finding may be related to several aspects. First, the temporary closure of third-level outpatient clinics over the pandemic period may have favored this dramatic increase. Patients with chronic and debilitating neurodegenerative diseases, including PD and MS, experienced a general worsening of their conditions during the COVID-19 spread, thus requiring more careful medical assistance [24,25,26]. Therapeutic management of MS and PD often requires specialist prescription and administration of drugs. These aspects may also explain the increased number of non-avoidable neurological evaluations during the lockdown compared with the pre-COVID-19 era. The rise of visits requesting a change in diagnosis, therapeutic management, or workup during the lockdown may be related to the more complex cases evaluated, reflecting the higher proportion of patients requiring hospitalization.

The differences in reasons for requesting a neurological evaluation were confirmed by the different prevalence of diagnoses at the discharge, showing a decrease in diagnosis of CVDs and primary headache and an increase in the diagnosis of multiple sclerosis and PD and movement disorder. However, the most relevant finding is the dramatic increase in diagnoses of NPSs, including depression, anxiety, insomnia, and psychosis. Neuropsychiatric disturbances represented the most frequent diagnosis during the lockdown, involving 18% of the non-deferrable neurological evaluations over October 2020–March 2021. The negative effect of the COVID-19 pandemic on the psychological status has been reported previously in the general population [27,28,29]. Social isolation and quarantine have been associated with a high prevalence of anxiety and depression [30]. Specific populations or groups of patients have been more prone to developing NPSs during the pandemic. Anxiety, depression, insomnia, and somatic symptoms significantly affected healthcare workers, who were exceptionally overworked during the spread of COVID-19 [31,32]. Practitioners working in healthcare developed anxiety, depression, and insomnia symptoms, particularly in the case of pre-existing chronic medical disorders such as autoimmune diseases [33]. Similarly, patients with concomitant chronic diseases have a higher risk of developing insomnia, anxiety, and depression [34]. Patients affected by neurological disorders are particularly exposed to the psychological consequences of social distancing and isolation and quarantine measures [5,35,36]. Here, we show an overall increased prevalence of neuropsychiatric disturbances in people requesting a non-deferrable neurological evaluation and a specific rise of diagnosis of depression, with a consequent higher number of prescribed antidepressants. Since the number of patients referred to the neurologist due to a neuropsychiatric symptom was similar in the two groups (the reason for requesting a non-deferrable neurological evaluation due to NPSs represented 3% of visits in the pre-COVID-19 group and 4% in the LOCKDOWN group), we can infer that most cases represented new formal diagnoses. Again, the severity of cases was likely higher in the lockdown than in the pre-COVID-19 era, given that antidepressants and benzodiazepines represented the most prescribed drugs (Figure 2). The higher number of patients suffering from NPSs may be due to several reasons, including fear of being infected, uncertainty about the future, disinformation, and economic concerns [9,37]. Long-term longitudinal studies will provide the exact severity of the phenomena, but it is already evident that a more careful attitude to preventing the psychological consequences of the pandemic-related changes is needed.

Demographics, clinical status, and comorbidities are essential aspects to be considered in work exploring epidemiological aspects during the pandemic [38]. Since comorbidities may worsen or provoke neurological disorders [39,40,41], we investigated the prevalence of comorbidities in patients accessing the non-deferrable neurology outpatient clinic. The three most prevalent comorbidities were hypertension, hypercholesterolemia, and diabetes in both groups, without significant differences between the pre-COVID era and during the lockdown. Likewise, no differences were found between groups regarding age and sex of patients, nor regarding the number of patients accessing the neurological evaluation for the first time. Unfortunately, other factors, such as habits, educational level, social status, and the economic level of patients could not be analyzed because they were not available. 

Our study has some limitations due to the retrospective design of the work and its possible missing data, including the presence of previous COVID-19 infection in patients evaluated during the lockdown. In addition, we could not explore the role of other variables of potential interest (e.g., socio-economic status) that were not collected in most cases of neurological non-deferrable visits. Lastly, this study involved a single neurology outpatient clinic. Thus, regional healthcare system organization should be considered in the interpretation of the present results.

Despite these limitations, our study shows that the COVID-19 pandemic induced several changes in patients’ management and access to the neurology unit. First, our data confirm a dramatic increase in NPSs in neurologic patients, suggesting that dedicated psychological assistance is needed to prevent mental health overload. In addition, the evidence of increased appropriateness in neurological non-deferrable evaluations during the lockdown bears witness to the fact that a rearrangement of the healthcare system is required, especially during non-pandemic periods.

## Figures and Tables

**Figure 1 jcm-10-05169-f001:**
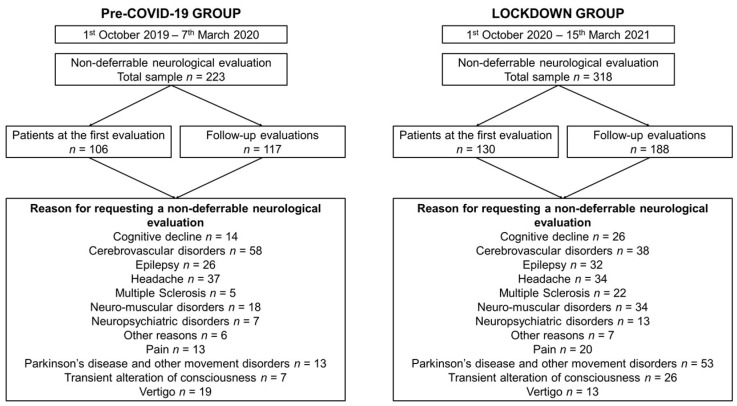
Flow-chart detailing sample selection, patient admission and reasons for requesting a non-deferrable neurological evaluation. Follow-up evaluations indicate patients previously evaluated at the same neurology outpatient clinic.

**Figure 2 jcm-10-05169-f002:**
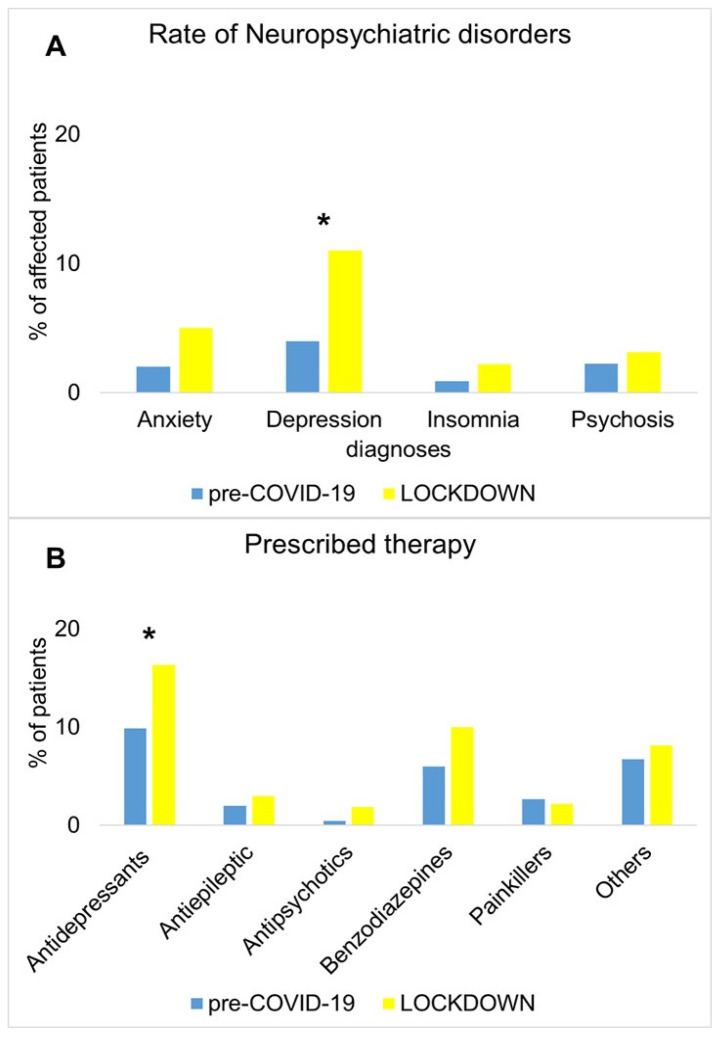
Differences in clinical diagnoses of neuropsychiatric disorders (panel **A**) and in prescribed therapy (panel **B**) between the pre-COVID-19 group (blue bars) and the LOCKDOWN group (yellow bars). * *p* < 0.05 was considered statistically significant.

**Table 1 jcm-10-05169-t001:** Patients’ demographics, reasons for requesting a non-deferrable neurological evaluation, and discharge diagnoses.

Group	Pre-COVID-19 (*n* = 223)	LOCKDOWN (*n* = 318)	*p* Value
**Age mean ± SD (years)**	62.8 ± 17.4	60.6 ± 17.3	0.154
**Sex female, N (%)**	111 (50%)	170 (53%)	0.399
**First evaluation N (%)**	106 (48%)	130 (41%)	0.125
**Reason for requesting a non-deferrable neurological evaluation N (%)**			
**Cognitive decline**	14 (6%)	26 (8%)	0.406
**CVDs**	58 (26%)	38 (12%)	0.000 **
**Epilepsy**	26 (12%)	32 (10%)	0.555
**Headache**	37 (17%)	34 (11%)	0.045 *
**MS**	5 (2%)	22 (7%)	0.014 *
**NM disorders**	18 (8%)	34 (11%)	0.309
**NPSs**	7 (3%)	13 (4%)	0.565
**Other**	6 (3%)	7 (2%)	0.714
**Pain**	13 (6%)	20 (6%)	0.826
**PD and other MDs**	13 (6%)	53 (17%)	0.000 **
**TAC**	7 (3%)	26 (8%)	0.016 *
**Vertigo**	19 (8%)	13 (4%)	0.031 *
**Prevalence of diagnosis at discharge N (%)**			
**Cognitive decline**	13 (6%)	20 (6%)	0.826
**CVDs**	49 (22%)	31 (10%)	0.000 **
**Epilepsy**	29 (13%)	36 (11%)	0.553
**Primary headache**	35 (16%)	28 (9%)	0.014 *
**MS**	3 (1%)	20 (6%)	0.005 **
**NM disorders**	29 (13%)	44 (14%)	0.780
**No diagnosis**	14 (6%)	12 (4%)	0.180
**Non-neurological**	12 (5%)	17 (5%)	0.986
**NPSs**	16 (7%)	57 (18%)	0.000 **
**Other**	10 (4%)	15 (5%)	0.899
**PD, ET and other MDs**	13 (6%)	38 (12%)	0.016 *

Abbreviations: CVDs: cerebrovascular disorders; ET: essential tremor; MD: movement disorders; N: number; NM: neuro-muscular; NPSs: neuropsychiatric disorders; PD: Parkinson’s disease; SD: Standard deviation; TAC: transient alteration of consciousness. * *p* < 0.05; ** *p* < 0.01.

## Data Availability

All data reported and discussed in this study are available from the corresponding author, upon reasonable request.

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
