# Peer review of "Increased Prevalence of Neuropsychiatric Disorders during COVID-19 Pandemic in People Needing a Non-Deferrable Neurological Evaluation"

_jcm, 2021, doi:10.3390/jcm10215169_

Round 1
Reviewer 1 Report
Dear Authors,
Thank you for considering submitting your manuscript to our journal. The topic of your manuscript is interesting and new, the investigated group of patients is well defined. Statistical methods are adequate and presentation of results is clear. It will contribute to getting more knowledge related to consequences of Covid-19.
The main question addressed in this research is increased rate of neuropsychiatric disorders during Covid-19 pandemic in people needing a non-deferrable neurological evaluation, what is relevant and interesting in the times of Covid-19 pandemic. This is original topic, as Covid-19 pandemic is the first time present in the world. It adds original data obtained in the group of people investigated. The paper is well-written and the text is clear and easy to read. The conclusions are consistent with the evidence presented. The authors are aware of some limitations of their study due to the retrospective design of their work and possible missing data and in addition, the study was carried on in a single Neurological Outpatient Clinic.
Author Response
Reviewer: 1.
The topic of your manuscript is interesting and new, the investigated group of patients is well defined. Statistical methods are adequate and presentation of results is clear. It will contribute to getting more knowledge related to consequences of Covid-19…
Reply to Reviewer 1. We thank the Reviewer for the positive comment.
Reviewer 2 Report
Dear Authors,
This communication is well written and meets the criteria for publication in JCM. It brings new knowledge to the extremely important issue of neurological and neuropsychiatric problems in the era of the COVID-19 pandemic. Nevertheless, there is no current reference to the impact of the COVID-19 pandemic on specific populations, including autoimmune disease patient populations. I recommend supplementing the communication with the following items:
Wańkowicz, P.; Szylińska, A.; Rotter, I. The Impact of the COVID-19 Pandemic on Psychological Health and Insomnia among People with Chronic Diseases. J. Clin. Med. 2021, 10, 1206. https://doi.org/10.3390/jcm10061206
Wańkowicz, P.; Szylińska, A.; Rotter, I. Insomnia, Anxiety, and Depression Symptoms during the COVID-19 Pandemic May Depend on the Pre-Existent Health Status Rather than the Profession. Brain Sci. 2021, 11, 1001. https://doi.org/10.3390/brainsci11081001
Best regards
Author Response
Reviewer: 2.
This communication is well written and meets the criteria for publication in JCM. It brings new knowledge to the extremely important issue of neurological and neuropsychiatric problems in the era of the COVID-19 pandemic. Nevertheless, there is no current reference to the impact of the COVID-19 pandemic on specific populations, including autoimmune disease patient populations. I recommend supplementing the communication with the following items…
Reply to Reviewer 2. We thank the Reviewer for the positive comment. We fully considered the suggestion to discuss the suggested references, and we added a novel brief section in the Discussion (page 7, line 250).
Reviewer 3 Report
This manuscript presents results from a retrospective observational study assessing non-deferrable neurological outpatients before the pandemic and during the Italian second wave of the COVID-19 pandemic. While the manuscript is potentially of interest, there are some important methodological flaws as outlined below:
- In introduction, please change “chronic brain disorders” to chronic neurological disorders.
- How were non-deferrable neurological visits defined and extracted from the electronic health records? The authors mention a scoring system used to characterize a visit as non-deferrable in retrospect; however, in methods it should be stated clearly whether all outpatient visits within these periods were analysed?
- Why does the pre-COVID-19 group contain March 2020 (in contrast to what the authors state in the Introduction concerning the begin of the pandemic in Italy)?
- At page 2, the sentence “A total of 223 visits in the pre-COVID-19 group and 318 in the LOCKDOWN group were analyzed” should be moved to the Results section.
- Also, how did the authors evaluate the baseline diagnoses (i.e., diagnoses prior to the outpatient visit) to assess and score the presence of “new diagnoses”? I would recommend presenting baseline diagnoses and distinguishing them from “new diagnoses”.
- The term “rate of diagnosis” is neither correct nor accurate and should be changed in Table 2, in the Results section and throughout the manuscript. Do the authors refer to new diagnoses or to prevalence? See also previous comment.
- For the results presented in Section 3.4 the authors have not reported the statistical test used for comparisons, or whether correction for multiple comparisons was performed. Similarly, the subanalysis presented in Section 3.5 has not been introduced in the Methods.
- Why were primarily neurologists involved in assessment of psychiatric conditions?
- Why was a score of at least two deemed as appropriate visit? I cannot exactly follow the rationale, the cut-off here seems a bit arbitrary.
- Some of the conclusions of the discussion such as "most severe cases were admitted during the lockdown, and more complex management was required." are not supported by the results of the manuscript.
Author Response
Reviewer: 3.
This manuscript presents results from a retrospective observational study assessing non-deferrable neurological outpatients before the pandemic and during the Italian second wave of the COVID-19 pandemic. While the manuscript is potentially of interest, there are some important methodological flaws as outlined below:
Reply to reviewer 3. We thank the Reviewer for the comments and suggestions. We fully considered the raised points as follows.
1. In introduction, please change “chronic brain disorders” to chronic neurological disorders.
Thanks, we made the correction.
2. How were non-deferrable neurological visits defined and extracted from the electronic health records? The authors mention a scoring system used to characterize a visit as non-deferrable in retrospect; however, in methods it should be stated clearly whether all outpatient visits within these periods were analysed?
We would like to thank the reviewer (and the editors) for the comment, we agreed that further clarification was needed and we made some changes to the scoring system. Specifically, we evaluated appropriateness of the neurological evaluations based on a previously published classification system, considering avoidable and unavoidable visits (Ref 17, Avoidable emergency department visits: a starting point., DOI: 10.1093/intqhc/mzx081). We performed a new statistical analysis based on the adopted classification system confirming a statistically significant difference between the two groups (page 5, line 170). We underlined in yellow all changes regarding the classification system (page 2, line 69; page 3, line 100; page 5, line 168).
3. Why does the pre-COVID-19 group contain March 2020 (in contrast to what the authors state in the Introduction concerning the begin of the pandemic in Italy)?
Thanks for the comment, we made some clarifications. We included only visits before the Italian national lockdown declared on the 9 th of March 2020 (as we specified in the introduction, page 2, line 51), and we amended the inclusion date, since the last included visit was performed on the 7 th of March, 2020.
4. At page 2, the sentence “A total of 223 visits in the pre-COVID-19 group and 318 in the LOCKDOWN group were analyzed” should be moved to the Results section.
Thanks, we moved the sentence as suggested (page 3, line 126).
5. Also, how did the authors evaluate the baseline diagnoses (i.e., diagnoses prior to the outpatient visit) to assess and score the presence of “new diagnoses”? I would recommend presenting baseline diagnoses and distinguishing them from “new diagnoses”.
Thanks for the comment. We agreed that the previous classification system could be misleading. As stated in reply to point 2, we modified the classification according to a previously published system classifying visits as “avoidable” (requiring no change in the diagnostic workup, further assessment, or medications at the discharge) and non-avoidable. Following the classification system, we did not consider “new diagnoses” as a criterion for evaluating the appropriateness of neurological evaluation. However, all diagnoses are reported in table 1 and described in the result section.
6. The term “rate of diagnosis” is neither correct nor accurate and should be changed in Table 2, in the Results section and throughout the manuscript. Do the authors refer to new diagnoses or to prevalence? See also previous comment.
Thanks for the comment, we clarified we were referring to prevalence of each diagnosis.
7. For the results presented in Section 3.4 the authors have not reported the statistical test used for comparisons, or whether correction for multiple comparisons was performed. Similarly, the subanalysis presented in Section 3.5 has not been introduced in the Methods.
Thanks for the comment, we specified the statistics and we introduced the comparison of neuropsychiatric symptoms and prescribed therapy in the methods section (page 3, line 120).
8. Why were primarily neurologists involved in assessment of psychiatric conditions?
This is an interesting point. Our study shows that the diagnoses of neuropsychiatric disturbances increased over the lockdown period compared with the pre-covid era. However, the reason for requesting a non-deferrable neurological evaluation due to neuropsychiatric disturbances represented only a negligible percentage of the visits in both the pre-COVID-19 and the LOCKDOWN group (3% and 4%, respectively). This result suggests that a novel diagnosis of neuropsychiatric disorders has been made during the lockdown (these findings are discussed on page 7, line 261).
9. Why was a score of at least two deemed as appropriate visit? I cannot exactly follow the rationale, the cut-off here seems a bit arbitrary.
Thanks for the comment. As stated in reply to point 2 and to point 5, we agreed that the previous classification system could be misleading, and we referred to a previously published classification system. Therefore, we amended figure 1 accordingly.
10. Some of the conclusions of the discussion such as "most severe cases were admitted during the lockdown, and more complex management was required." are not supported by the results of the manuscript.
Thanks for the remark. We amended all the marked sentences to provide a more detailed discussion of the results.
Round 2
Reviewer 3 Report
The authors have sufficiently addressed all concerns raised during the review process, I have no further comments.